# MulTNet: A Multi-Scale Transformer Network for Marine Image Segmentation toward Fishing

**DOI:** 10.3390/s22197224

**Published:** 2022-09-23

**Authors:** Xi Xu, Yi Qin, Dejun Xi, Ruotong Ming, Jie Xia

**Affiliations:** State Key Laboratory of Mechanical Transmission, Chongqing University, Chongqing 400044, China

**Keywords:** multi-scale transformer, contextural information, semantic segmentation, marine animal, subregion

## Abstract

Image segmentation plays an important role in the sensing systems of autonomous underwater vehicles for fishing. Via accurately perceiving the marine organisms and surrounding environment, the automatic catch of marine products can be implemented. However, existing segmentation methods cannot precisely segment marine animals due to the low quality and complex shapes of collected marine images in the underwater situation. A novel multi-scale transformer network (MulTNet) is proposed for improving the segmentation accuracy of marine animals, and it simultaneously possesses the merits of a convolutional neural network (CNN) and a transformer. To alleviate the computational burden of the proposed network, a dimensionality reduction CNN module (DRCM) based on progressive downsampling is first designed to fully extract the low-level features, and then they are fed into a proposed multi-scale transformer module (MTM). For capturing the rich contextural information from different subregions and scales, four parallel small-scale encoder layers with different heads are constructed, and then they are combined with a large-scale transformer layer to form a multi-scale transformer module. The comparative results demonstrate MulTNet outperforms the existing advanced image segmentation networks, with *MIOU* improvements of 0.76% in the marine animal dataset and 0.29% in the ISIC 2018 dataset. Consequently, the proposed method has important application value for segmenting underwater images.

## 1. Introduction

With the development of marine ranching, there are serious drawbacks to traditional manual fishing, including the high risk of fishing, danger of fishermen, low efficiency, and small working area. Therefore, it is worth researching the autonomous underwater vehicle (AUV). The object segmentation plays an important role in the AUV. In recent years, machine vision technology has been widely used for target detection and segmentation owing to the advantages of low cost and high precision. Some methods based on deep learning [1,2,3] have been proposed for the detection and recognition of marine organisms. However, these methods based on convolutional neural networks (CNNs) mainly focus on image classification tasks, and they cannot predict the specific shapes and sizes of marine organisms, which affects the accurate grab of marine animals by the robotic gripper. Consequently, it is extremely significant for implementing accurate marine image segmentation.

As an important branch of computer vision, image semantic segmentation can perform pixel-level classification by labelling each pixel of an image into a corresponding class, which is also known as dense prediction. Semantic segmentation is suitable for a variety of tasks, including autonomous vehicles and medical image diagnostics. However, there are few studies on marine animal segmentation. In addition, due to the noise caused by flowing particles, light scattering, and a complex underwater background, the acquired marine images are generally vague or of poor quality, which seriously affects the accuracy of marine image segmentation. To overcome these issues mentioned above, this work aims to explore a new transformer-based image segmentation method for improving the precision of marine animal segmentation and meeting the requirements of automatic fishing.

In the past decade, deep learning has become a hot topic in image segmentation. In particular, various CNNs have been widely applied to semantic segmentation. Although the CNN network does well in extracting the low-level features, it is difficult to capture the global semantic information owing to the limitation of the receptive fields of convolution kernels. On the other hand, as a new technique of deep learning, the transformer has excellent performance in dealing with high-level semantic information, which can effectively capture the long-distance dependence. However, since the patch size is fixed, it is hard for the transformer to acquire the low-resolution and multi-scale feature maps, which brings great difficulties to the segmentation task. Whereupon we developed a new multi-scale transformer network for semantic segmentation that possesses the advantages of both CNN in extracting the low-level features and transformer in processing the relationship between two visual elements, and it is named MulTNet. The MulTNet mainly consists of a dimensionality reduction CNN module, a multi-scale transformer module, and a decoder. With the principle of progressive downsampling, the dimensionality reduction module is designed. It first uses two convolution layers to gradually decrease the resolution of feature maps while flexibly increasing the number of channels, and the feature maps with sufficient low-level semantic information are obtained, then they are flattened into a low dimension matrix as the input of the multi-scale transformer module. The multi-scale transformer module is constructed through four parallel small-scale encoder layers with different heads and a conventional large-scale encoder layer. In this module, the four small-scale encoder layers can help to extract the multi-scale contextural features and improve the computational efficiency through parallel computing, while the large-scale transformer encoder layer can further extract the high-level features on the basis of multi-scale features. Finally, several CNN-based decoder layers are used to recover the original image resolution for better segmentation results. Compared with other existing typical segmentation methods, MulTNet demonstrates higher segmentation performance in the marine animal dataset and the public ISIC 2018 dataset. Therefore, it has great application potential in marine fishing. The contributions of this paper are summarized as:

(1) A new semantic segmentation network named MulTNet is proposed, and it can extract the low-level features and the multi-scale high-level semantic features, thereby improving the segmentation accuracy. The proposed network is suitable for various object segmentation tasks, especially for marine animal segmentation;

(2) A new multi-scale transformer module has been developed. The four transformer layers with different heads can parallel extract the contextural characteristics in different subspaces and scales, which enhances the feature expression ability of the proposed network and increases the computation speed;

(3) A dimensionality reduction CNN module based on progressive downsampling is designed for extracting the low-level features. Via the flatten operation, the obtained low-dimension matrix can alleviate the computational burden of the subsequent transformer. Additionally, the ablation experiment verifies the effectiveness of this module.

The remainder of this paper is organized as follows. A summary of the related work is reviewed in Section 2 and the overall architecture of our proposed method is described in detail in Section 3. Section 4 presents the training details and discusses the results with comparisons to state-of-the-art methods. Section 5 focuses on conclusions and the prospect of future work. 

## 2. Related Work

**Semantic segmentation:** As one of the earliest segmentation networks based on the encoder-decoder structure, FCN [4] designed a skip architecture to combine the deep and coarse information with the shallow and fine information to improve the segmentation accuracy. However, FCN does not consider global context information, and its predictions are relatively rough [5]. As one of the most popular methods, U-Net [6] can utilize the global location and context simultaneously, which has good performance in image segmentation [7]. As an extension to U-Net, Attention U-Net [8] showed great improvements in the accuracy and sensitivity of the foreground pixels by integrating a self-attention gating module into the conventional CNN-based segmentation network. Lin et al. [9] developed RefineNet to learn the high-level semantic features by adding residual connections. The authors of [10] used ResUNet take advantage of both the U-Net network and deep residual learning and promoted information propagation by concatenating the low-level details and the high-level semantic features. 

Furthermore, research by [11,12,13] addressed expanding receptive fields and improving multi-scale information for semantic segmentation. Yu et al. [14] and Chen et al. [15] put forward dilated or atrous convolutions. The authors of [16] made use of Deeplabv3+ and used atrous separable convolution in both the decoder module and the ASPP model to probe different features from different scales and spaces by applying depth-dependent separable convolution to ASPP. On the basis of dilated residual networks, the authors of [17] utilized PSPNet and designed the PPM module to enhance the extraction ability of context information from different subspaces. As a cascading knowledge diffusion network, the authors of [18] utilized CKDNet and boosted segmentation performance by fusing information learnt from different tasks. Although the above approaches have great success in enriching the spatial context, it is still hard to capture the long-range dependencies and global information.

**Underwater segmentation:** Although the current segmentation methods show excellent performance on various tasks, such as medical image segmentation and autonomous vehicle image segmentation, they face a challenge for the accurate segmentation of underwater images. Hence, some specific methods were proposed to segment the marine animal images. Wei et al. [19] designed the Unet-FS network to segment different types of free surfaces. Yao et al. [20] proposed a new segmentation method for fish images, and it used the mathematical morphology and K-means clustering segmentation algorithm to separate the fish from the background. The authors of [21,22,23] explored several non-End2End segmentation architectures to obtain high accuracy on the seagrass images. Furthermore, a number of underwater image enhancement algorithms [24,25,26,27,28] have been proposed, which can be beneficial to the subsequent segmentation. If orthogonal polynomials are used [29,30], the accuracy of segmentation can be further improved. To improve the quality of images, some advanced underwater image sensors have been proposed [31,32,33]. Several publicly available underwater datasets [34,35,36] have been published to test the performance of various image processing approaches. In this paper, we focus on researching a new marine animal segmentation method to enhance the ability of automatic fishing.

**Transformer:** With the self-attention mechanism and Seq2Seq structure, the transformer was originally designed for processing natural language. Vaswani et al. [37] first proposed a transformer for machine translation and achieved good results. Devlin et al. [38] proposed a new language representation model called BERT to pretrain a transformer based on the unlabeled text considering the content of each word. BERT demonstrated excellent performance on eleven NLP tasks. Other transformer variants have also been proposed for solving NLP tasks, such as the famous BART [39], GPT-3 [40], and T5 [41]. Recently, due to the strong representation capabilities of transformers, they have been widely used for the various vision tasks, including semantic segmentation [42,43], object detection [44,45], and image recognition [46]. Dosovitskiy et al. [47] proposed a vision transformer (ViT), and it possessed the satisfactory performance of image classification owing to the large-scale pretraining of a convolution-free transformer. As ViT requires a large amount of pretraining data, Touvron et al. [48] developed a data-efficient image transformer (DeiT), and it utilized a token-based distillation strategy and teacher model to guide the better learning of DeiT. A swin transformer [49] was built as a vision backbone using hierarchical architecture and shifted window operation.

Although these transformer-based methods have achieved great success in image classification tasks, they rarely obtain the feature maps with different scales, which affects their segmentation capacities. Thereupon, some hybrid models combining CNNs, and transformers were proposed, especially for small-scale datasets. Chen et al. put forward TransUnet for medical image segmentation via combining transformer with U-Net. Xie et al. [50] proposed a hybrid network (CoTr) that effectively bridged CNNs and transformers, achieving a significant performance improvement for 3D medical image segmentation. Incorporating with a transformer, TrSeg [51] adaptively produced multi-scale contextual information with dependencies on original contextual information. As a task-oriented model based on skin lesion segmentation, FAT-Net [52] increased the receptive field for extracting more global context information.

Unfortunately, the above hybrid networks just use the transformers but do not improve the feature extraction ability of the transformer. Meanwhile, the transformer usually has a large computational burden. To address the above issues, a multi-scale transformer is explored in this work, and it can not only effectively extract the multi-scale features but also increase the computing speed. MulTNet, which integrates CNN and the multi-scale transformer, is also proposed to combine the ability of CNN in capturing local visual perception with the capacity of the transformer in building long-range dependencies. The low-level features and the multi-scale high-level semantic features can finely represent the characteristic information from various objects and noise. The proposed network has powerful segmentation performance, and it is superior to the current classical segmentation methods, which is validated by two datasets. All the segmentation method mentioned in the literature are listed in Table 1.

## 3. Proposed Network

The proposed MulTNet has a network architecture of encoder-decoder. To reduce the computational burden, a dimensionality reduction module based on CNN is designed, which consists of two convolution layers and a flattening layer. Two convolutional layers can effectively reduce the dimensions of feature maps, which is helpful for simplifying the tensor operation and reducing the memory requirement. Then, we propose a multi-scale transformer module as the encoder of MulTNet. The encoder consists of four parallel small-scale transformer encoder layers and one conventional large-scale transformer encoder layer. Four small-scale transformer encoder layers with different heads can extract useful information from different scales, and their outputs are combined into the input of the large-scale conventional transformer layer for further feature extraction. Via combining both transformer and CNN branches in our network, not only can the abundant local features be extracted, but also the important global context information can be fully captured for marine animal segmentation. The decoder uses step-by-step convolutions to implement the restoration of the segmented image. The detailed structure diagram of MulTNet is shown in Figure 1.

### 3.1. Dimensionality Reduction CNN Module

In most current vision-transformer networks, the convolution operation is adopted for downsampling, that is, the front-end high-dimensional image is transformed into a low-dimensional tensor. As previously mentioned, a hybrid model combining transformer with CNN always shows better performance than a single transformer. Furthermore, convolutional layers generate the feature maps pixel by pixel [53], which contributes to an excellent performance in capturing local features. It then follows that step-by-step downsampling is employed to reduce the resolution of input images. Simultaneously, the channels in each CNN layer are gradually expanded. In this way, the local information of the input image can be extracted as much as possible. Specifically, we first reduce the resolution and expand the channels of the original image $x\in R^{H\times W\times 3}$ is converted into $X\in R^{H/16\times W/16\times C}$ by adopting two convolution layers, where *H, W,* and *C* denote the height, width, and channels of the feature maps respectively. One layer has 192 convolution kernels with a size of (8×8) and a stride of 8, while the other layer has 768 convolution kernels with a size of (2×2) and a stride of 2. The two convolution operations can be expressed as:(1)$$fm_{1}=Conv_{8\times 8\times 192}(RELU(BatchNorm(X_{input})))$$
(2)$$fm_{2}=Conv_{2\times 2\times 768}(RELU(BatchNorm(fm_{1})))$$
where *fm*_1_ and *fm*_2_, respectively, denote the output feature maps of the first layer and the second layer.

Considering that the transformer just accepts the 1D sequence of patch embeddings as the input, every single feature map is flattened into a 1D sequence. In order to handle different patch sizes, the linear projection is used to map the patch embeddings into the vectors with a fixed length, and they form a C-dimensional embedding space $E\in R^{L\times C}$, where *L* is the length of a single sequence, which is set to *HW*/256, and *C* is the number of channels. To keep the spatial position information of patches, the single position embedding is added with the patch embedding to obtain the final input of the multi-scale transformer module. This computational procedure is formulated as:(3)$$E_{t}=\left\{p_{1}+e_{1},p_{2}+e_{2},\ldots ,p_{C}+e_{C}\right\}$$
where *p**i* and *e**i**,* respectively, denote the single position embedding and the 1D patch embedding in the *i*-th channel; $E_{t}$ represents the final linear flattened 2D embedding space.

### 3.2. Multi-Scale Transformer Module

Without the capability of building long-range relationships, CNNs cannot precisely distinguish the differences between pixels for some challenging work, including underwater image segmentation, where the problems of insufficient resolution, low contrast, and blurred marine animal boundaries are usually present. Unlike the convolutional layers, which gain global features only by stacking more layers, the transformer can effectively build the long-range relationships to capture the rich global context information. Inspired by the multi-scale strategy in deep learning [54], we employ a multi-scale transformer module to capture global features from different subregions and scales. A conventional transformer struggles to capture contextural information in different subregions and scales with a fixed number of heads and too many patches for training, and its computational efficiency is low. The 2D embedding space, $E_{t}$ obtained by the dimension reduction CNN module is then divided evenly into four small embedding matrices, $E_{s}\in R^{L\times C/4}$ which are fed into four parallel transformer encoder layers with different heads. 

As the core of the transformer, multi-head attention is proposed via the stacking of multiple simple self-attention functions, which helps to capture different spatial and semantic information from different subspaces. The output of a simple self-attention module can be written as:(4)$$ATT(Q,K,V)=Softmax(\frac{QK^{T}}{\sqrt{d_{K}}})V$$
where *Q, K,* and *V* represents query, key, and value matrices respectively; $d_{K}$ denotes the dimension of *Q* and *K*. The multi-head attention linearly projects *Q*, *K,* and *V* into *h* different subspaces via the self-attention operation, and then the outputs of multiple heads (self-attention modules) are concatenated. The calculation process of multi-head attention is formulated as:(5)$$H_{i}=ATT(QW_{i}^{Q},KW_{i}^{K},VW_{i}^{V})$$
(6)$$ATT_{MultiHead}(Q,K,V)=Concat(H_{1},\ldots ,H_{h})W^{o}$$
where $QW_{i}^{K}$, $KW_{i}^{K}$, and $VW_{i}^{V}$ respectively represent the query, key, and value of the *i*-th head, and $W_{i}^{Q},W_{i}^{K}\in R^{D_{model}\times d_{k}}$ and $W_{i}^{V}\in R^{D_{model}\times d_{v}}$ are the corresponding weight matrices, in which $D_{model}$ is the dimension of embedding and $d_{v}$ denotes the dimension of *Q*. $W^{O}\in R^{hd_{v}\times D_{model}}$ denotes the weight matrix.

In our study, four transformer encoder layers with 1, 2, 3, and 4 heads, respectively, are designed to capture more contextural information from different scales and different subregions. Through a series of tests, we find that the semantic features at different scales can be well extracted according to the designed head numbers. Each transformer layer takes a small embedding matrix (256×192) as the input, and the four transformer layers can be formulated as:(7)$$T_{s\_1}=ATT^{1}{}_{MultiHead}(Q^{1},K^{1},V^{1})=ATT(QW_{1}{}^{Q},KW_{1}{}^{K},VW_{1}{}^{V})N^{1}$$
(8)$$T_{s\_2}=ATT^{2}{}_{MultiHead}(Q^{2},K^{2},V^{2})=Concat(H_{1}{}^{2},H_{2}{}^{2})N^{2}$$
(9)$$T_{s\_3}=ATT^{3}{}_{MultiHead}(Q^{3},K^{3},V^{3})=Concat(H_{1}{}^{3},H_{2}{}^{3},H_{3}{}^{3})N^{3}$$
(10)$$T_{s\_4}=ATT^{4}{}_{MultiHead}(Q^{4},K^{4},V^{4})=Concat(H_{1}{}^{4},H_{2}{}^{4},H_{3}{}^{4},H_{4}{}^{4})N^{4}$$
where $N^{1},N^{2},N^{3},N^{4}\in R^{(h*\frac{C}{4h})\times \frac{C}{4}}$ denote the weight matrices obtained in the first, second, third, and fourth transformer layers, respectively; $H_{j}^{i}\in R^{\frac{C}{4*i}}$ denotes the *j*-th head of the *i*th transformer layer. We concatenate all these outputs into a $R^{L\times C}$ matrix, which is written as:(11)$$MulT=Concat(T_{1},T_{2},T_{3},T_{4})$$

To better establish the long-distance dependencies among those results obtained by the four small-scale transformer encoder layers, we adopt another large-scale transformer encoder layer with 5 heads to further capture the global contextural characteristics, and the computational process is formulated as:(12)$$T_{output}=ATT_{MultiHead}(Q_{o},K_{o},V_{o})=Concat(H_{1},H_{2},H_{3},H_{4},H_{5})N^{o}$$
where $N^{O}\in R^{C\times C}$; $H_{i}\in R^{\frac{C}{5}}$ denotes the *i*-th head in the large-scale transformer obtained by the even division of *MulT*.

### 3.3. Loss of MulTNet

The categorical cross-entropy loss (CE loss) function is expressed as the difference between the true probability distribution and the predicted probability distribution. The smaller the value of cross entropy, the better the prediction effect of the model. The formula can be expressed as:(13)$$E=-\sum_{i=1}^{N}y_{i}*log(S(f(x_{i})))$$
where *N* denotes the number of categories; $y_{i}$ denotes the true target tensor for the *i*-th category of samples, while $f(x_{i})$ denotes the predicted target tensor, that is, the output of the decoder, as shown in Figure 1.; *S* represents the softmax function.

## 4. Experiments and Discussion

### 4.1. Dataset Description

**Marine animal dataset:** The underwater images come from a Real-world Underwater Image Enhancement (RUIE) dataset [55], which can be downloaded from https://github.com/dlut-dimt/Realworld-Underwater-Image-Enhancement-RUIE-Benchmark (accessed on 7 August 2021). RUIE is the first large underwater image dataset for validating various object detection algorithms. In this work, 1396 underwater images are selected for our segmentation task, which have four categories, including background, sea cucumber, sea urchin, and starfish. LabelMe that can label the images is utilized to obtain the ground truth of the dataset. Then the dataset is divided into the training set, validation set, and test set, which respectively have 1116, 140, and 140 samples according to the ratio of 8:1:1. These images have various light scattering effects for verifying the proposed segmentation method.

**ISIC dataset:** This dataset was published by the International Skin Imaging Collaboration (ISIC) in 2018. As a large-scale dataset of dermoscopic images, it contains 2594 images with their corresponding ground truth annotations, which are available at https://challenge2018.isic-archive.com/ (accessed on 19 October 2021). For the ISIC dataset, we used 2074 images as the training set, 260 images as the test set, and 260 images as the validation set.

### 4.2. Model Trainings

We train MulTNet using mini-batch stochastic gradient descent (SGD) with an initial learning rate of 0.03, batch size of 6, and momentum of 0.9. To reduce the training time, the multi-scale transformer module adopts parallel training. For the marine animal dataset, 400 epochs are implemented, while 200 epochs are implemented for the ISIC dataset. Other hyper-parameters for both datasets are listed in Table 2. Furthermore, the following software and hardware configurations were used in the experiments: TensorFlow2.4 + Windows10 + CUDA 10.2 + CUDNN7.6.5 + NVIDIA GeForce RTX 3080.

### 4.3. Evaluation Metrics

In order to quantitatively evaluate the segmentation capability of the proposed MulTNet, we use a variety of evaluation metrics, including *MIOU*, *IOU*, *Acc*, and *MPA* [56]. The confusion matrix between prediction and ground truth is calculated via the number of positive pixels for the true identification (TP), the number of negative pixels for the true identification (TN), the number of negative pixels for the false identification (FP), and the number of positive pixels for the false identification (FN). Assume that *i* represents the true value, *j* represents the predicted value, $p_{ij}$ denotes the number of pixels originally belonging to class *i* but predicted to be class *j*, and *k* + 1 denotes the number of categories (including background). 

*IOU* denotes the overlap rate of the predicted images and ground truths, which is the ratio between their intersection and union. The ideal situation is a complete overlap, namely, the ratio is equal to 1. *IOU* is defined as
(14)$$IOU=\frac{TP}{TP+FN+FP}$$

*MIOU* is obtained by calculating the average of *IOUs* for all categories, and the computational formula is written as:(15)$$MIOU=\frac{1}{k+1}\sum_{i=0}^{k}\frac{p_{ii}}{\sum_{j=0}^{k}p_{ij}+\sum_{j=0}^{k}\left(p_{ij}-p_{ii}\right)}$$

*Acc* represents the pixel accuracy rate, and it is calculated as follows:(16)$$PA=\frac{\sum_{i=0}^{k}p_{ii}}{\sum_{i=0}^{k}\sum_{j=0}^{k}p_{ij}}$$

*MPA* stands for mean pixel accuracy, which is defined as follows:(17)$$MPA=\frac{1}{k+1}\sum_{i=0}^{k}\frac{p_{ii}}{\sum_{j=0}^{k}p_{ij}}$$

### 4.4. Comparison with State-Of-The-Art Methods

Using the marine animal dataset and the ISIC dataset, the proposed MulTNet is compared with the state-of-the-art methods including FCN, U-Net, Attention U-Net, RefineNet, ResUnet, DeeplabV3+, CKDNet, PSPNet, TrSeg, SegFormer [57], and FAT-Net. The evaluation metrics defined in Section 4.3 are used for quantitatively analyzing the segmentation abilities of various networks. Meanwhile, for a fair comparison, the hyper-parameters of the comparative methods are the same as those of MulTNet, which are listed in Table 2.

Firstly, all the used segmentation networks are applied to the marine animal dataset, which has great difficulty in segmentation owing to the blurry images and irregular shapes of marine organisms in the complex marine environment. The evaluation metrics obtained by MulTNet and the comparative networks are listed in Table 3. It is obvious that the *MPA*, *MIOU*, and *Acc* of MulTNet are up to 47.69%, 45.63%, and 97.27%, respectively, which are 0.52%, 0.76%, and 0.11% higher than those of the second-ranked FAT-Net. In the meantime, they are 10.82%, 10.15%, and 0.22% higher than those of the lowest-ranked FCN. In addition, the proposed network has the largest *IOUs* for the objects of sea urchin and starfish, while the *IOU* of MulTNet is only smaller than that of PSPNet for the object of sea cucumber. Consequently, our proposed network has better segmentation performance for the marine animal dataset than other methods.

FCN, U-Net, and Attention U-Net cannot make accurate predictions of marine animals when the underwater images are mostly blurry and have low contrast because they lack strong extraction ability for global contextual information. By taking advantage of both the U-Net framework and residual units, ResUnet obtains better accuracy performance than RefineNet. Relying on atrous convolution to obtain a large receptive field and the ASPP module to encode multi-scale contextual information, Deeplabv3+ also has a higher performance than ResUNet. CKDNet exploits two novel features entangled to boost the performance of classification and fine-level segmentation, so it outperforms the Deeplabv3+ method. Compared to CKDNet, PSPNet utilizes the pyramid pooling module to extract global context information from different regions, which shows the improved segmentation performance. PSPNet generates multi-scale contextual information without any prior information, while TrSeg can better generate it by adapting a transformer architecture to build dependencies on original contextual information. Without the interpolation of positional codes leading to decreased performance, SegFormer employs a MLP decoder to aggregate features from different layers, hence it is superior to TrSeg. As a hybrid encoder-decoder network consisting of CNN and transformer encoder branches, Fat-Net uses a FAM module and fuses it into each skip connection for enhancing the feature fusion between two feature maps, which can enhance the segmentation performance. However, these methods cannot establish long-term relationships or extract global context information, preventing segmentation performance from being improved further. Our proposed method, which is based on DRCM and MTM, can fully capture both local features and establish long-range dependencies to improve segmentation performance.

To visualize the segmentation results obtained by various networks, several typical examples of segmentation results are shown in Figure 2. The first and second columns are the original images and ground truths, respectively, and the next columns are, respectively, the segmentation results of FCN, U-Net, Attention U-Net, RefineNet, ResUnet, DeeplabV3+, PSPNet, TrSeg, CKDNet, SegFormer, FAT-Net, and MulTNet. Figure 2 shows that the predicted shapes of sea urchins and starfish obtained by MulTNet are more similar to the ground truths, particularly for some irregular or inconspicuous contours. The intuitive comparison further demonstrates that MulTNet has a higher accuracy of segmentation.

In addition, several examples from different levels of blur are selected to demonstrate the segmentation ability of our proposed method. As shown in Figure 3, our proposed method can precisely segment the marine animal nearly without any visible defects in the blurred underwater situation, primarily because DRCM with a large number of channels can precisely capture abundant low-level information. Meanwhile, the MTM can establish the long-distance dependence between blurred pixels and fully extract the high-level contextural information from different subregions and scales.

Next, to further verify the superiority of MulTNet, all methods are applied to segment the public ISIC 2018 dataset, and Table 4 shows their evaluation results. Evidently, four evaluation metrics of the proposed MulTNet outperform those of the existing classical networks, and the *MPA*, *IOU*, *MIOU*, and Acc of MulTNet are up to 94.04%, 81.48%, 88.09%, and 95.71%, respectively. FAT-Net and CKDNet are specially designed for skin lesion segmentation. It then follows that MulTNet has stronger segmentation ability and higher pixel-level segmentation performance than other typical networks. Similarly, several typical segmentation examples obtained by all the methods are illustrated in Figure 4 for a visualized comparison. It can be noted from Figure 4. that MulTNet extracts more details related to the shapes, especially for the dermoscopic image shown in the last row. Therefore, the advantage of the proposed MulTNet is validated again.

As is well-known, the loss function is used for network training, and it can measure the prediction accuracy and convergence rate. With regard to the marine animal dataset and the ISIC 2018 dataset, the loss curves of all networks are respectively drawn in Figure 5a,b. In Figure 5a, MulTNet has a minimum converged loss of over 0.1% lower than that of second-ranked FAT-Net, which means our segmentation network can better learn the semantic features compared to other segmentation networks. Meanwhile, the convergence rate of our model is basically the same as Attention U-Net. From Figure 5b, it can be clearly observed that our proposed MulTNet has the fastest convergence rate, and its converged loss is approximately equal to that of ResUNet. Compared with other networks, our proposed method can be convergent with fewer epochs, which leads to a faster convergence speed. The above analysis shows that MulTNet has higher convergence performance than other typical segmentation networks for both datasets, so it has more powerful segmentation ability.

Finally, we compare the total parameters, training speed, and testing speed of each model, and the results are illustrated in Table 5. FCN-8s have the maximum number of parameters (134 M), which leads to the slowest training speed. However, not all of these parameters have a positive influence on the final segmentation performance. U-Net is a lightweight U-shaped network that has the minimum total parameters and the fastest training speed. Owing to the proposed dimensionality reduction CNN module and multi-scale transformer module, the training speed and testing speed of MulTNet are the second fastest, which only take 0.183 s per iteration and 0.087 s per image, respectively. Although the training speed and testing speed of our model are slightly slower than those of U-Net, the segmentation accuracy of our model is obviously higher than that of U-Net.

### 4.5. Ablation Experiment

To verify the effectiveness of each module of the proposed dimensionality reduction CNN module (DRCM) and multi-scale transformer module (MTM), the original transformer, DRCM-Transformer, and MulTNet are used in the ablation experiments. For the ISIC and marine animal datasets, the segmentation results obtained by three approaches are listed in Table 6. It can be seen from this table that the *MIOU* and *MPA* on the marine animal dataset obtained by the original transformer are just 23.99% and 25.00%, respectively, as it cannot capture the characteristics at different scales, which also represent the feature information from different objects and noise. Compared with the original transformer, the DRCM-Transformer extracts the low-level features, so it possesses stronger segmentation ability. By adding MTM into DRCM-Transformer, the high-level contextural characteristics in different subspaces and scales can be extracted, so MulTNet has higher *MIOU* and *MPA* than DRCM-Transformer. The ablation experiment results show that DRCM and MTM play important roles in the proposed framework, and they can enhance the segmentation performance of the original transformer.

## 5. Conclusions

Since the shapes of various marine animals are very different and the collected images are influenced by the complex and varying underwater environment, it is a great challenge to achieve precise marine image segmentation. To fill this gap, a new multi-scale transformer segmentation network named MulTNet is proposed. The MulTNet is composed of a dimensionality reduction CNN module, a multi-scale transformer module, and a decoder. The dimensionality reduction CNN module is designed for generating the low-level feature maps, and then they are fed into the multi-scale transformer module, which is the main contribution of this work. The proposed multi-scale transformer module utilizes four small-scale transformer layers with different heads to extract the contextual characteristics from different subregions and scales, while a large-scale transformer is designed for further extracting the high-level semantic features. In addition, the parallel computation of four small-scale layers can effectively increase the computation speed. The results of contrast experiments indicate that MulTNet has higher segmentation performance than the existing typical methods for both the marine animal dataset and the ISIC dataset. Therefore, the proposed segmentation network can be better applied to segment the marine animals. In future research, an effective image preprocessing method will be explored for enhancing the quality of the marine images, which can help improve the segmentation accuracy, and we will apply our method to the AUVs for automatic fishing.

## Figures and Tables

**Figure 1 sensors-22-07224-f001:**
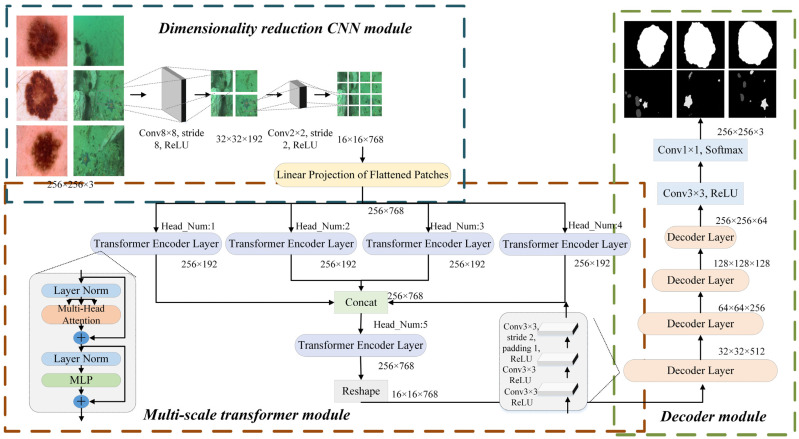
The detailed structure diagram of MulTNet.

**Figure 2 sensors-22-07224-f002:**
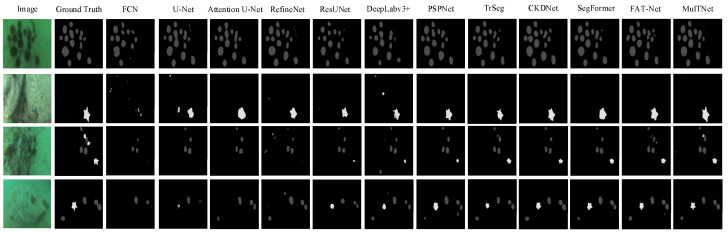
Examples of the marine animal segmentation obtained by FCN-8s, U-Net, Attention U-Net, RefineNet, ResUNet, Deeplabv3+, PSPNet, TrSeg, CKDNet, SegFormer, FAT-Net, and the proposed MulTNet.

**Figure 3 sensors-22-07224-f003:**
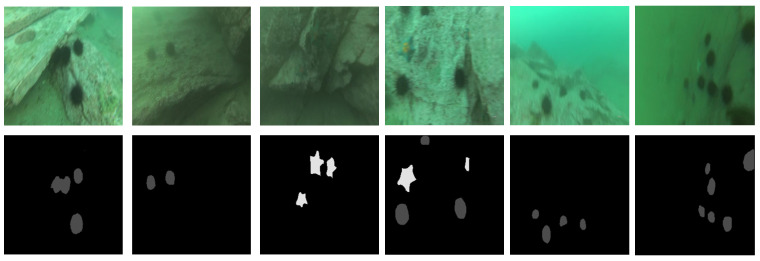
Examples of different levels of blur obtained by the proposed MulTNet.

**Figure 4 sensors-22-07224-f004:**
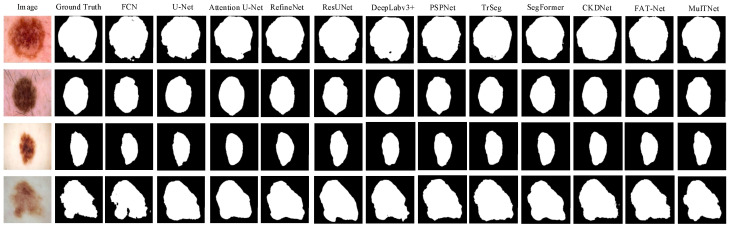
Examples of the marine animal segmentation obtained by FCN-8s, U-Net, Attention U-Net, RefineNet, ResUNet, Deeplabv3+, PSPNet, TrSeg, SegFormer, CKDNet, FAT-Net, and the proposed MulTNet.

**Figure 5 sensors-22-07224-f005:**
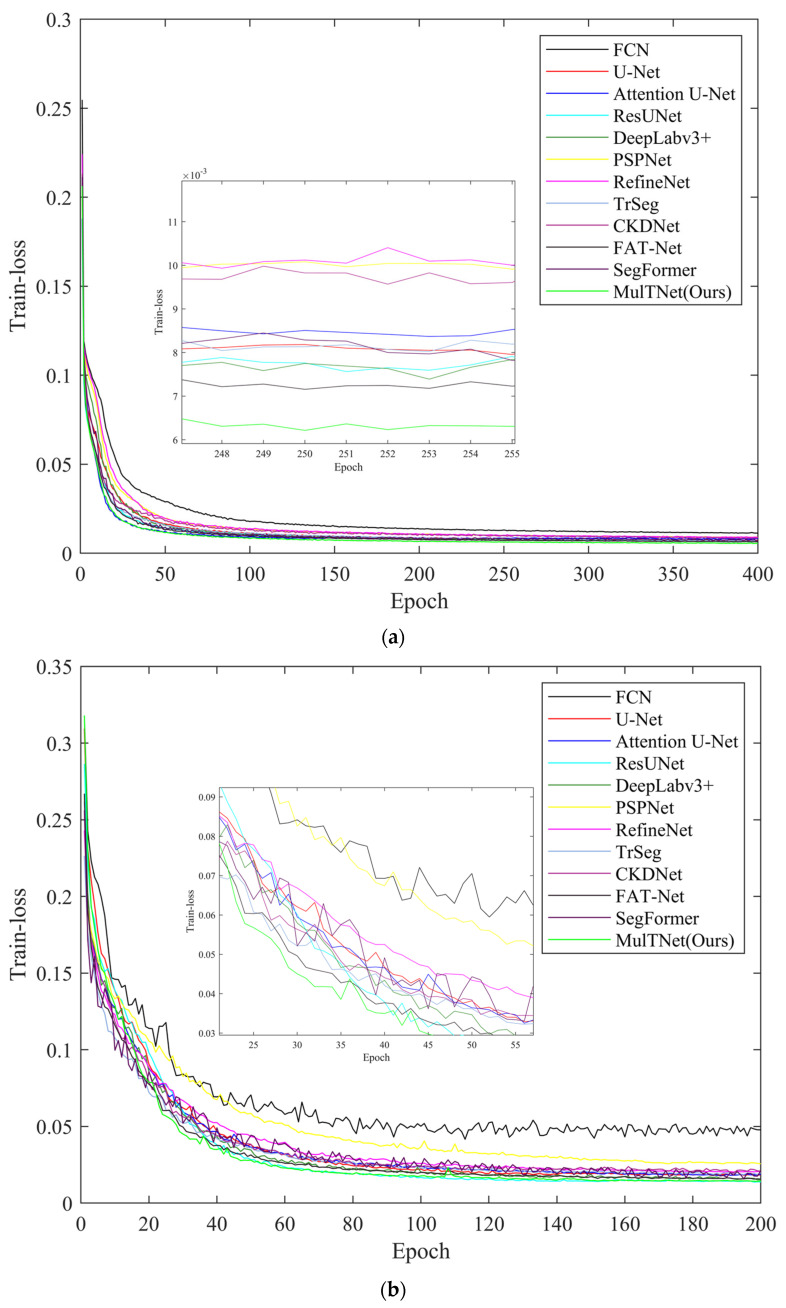
The training losses of various segmentation networks for two datasets: (**a**) dataset of marine animals; (**b**) dataset of ISIC 2018.

**Table 1 sensors-22-07224-t001:** A summary of the segmentation method mentioned in the literature.

Methodology	Year	Highlights	Limitations
FCN [4]: Fully convolutional network	2015	It is the first time CNN is used to extract features in the field of semantic segmentation.	This model has poor robustness on the image detail and does not consider the relationship between pixels.
U-Net [6]: U-shaped end-to-end network with skip connection.	2015	This model exploits skip connections between encoder and decoder to decrease the loss of context information.	This model generally shows poor performance in capturing details and explicitly building long-range dependency.
Attention U-Net [8]: Extension of standard U-Net model with attention mechanism.	2018	Suppress irrelevant areas in the input image and highlight the salient features of specific local areas.	Attention gates (AGS) mainly focus on extracting the spatial information of the region of interest, and the ability to extract less relevant local regions is poor.
RefineNet [9]: a generic multi-path refinement network.	2016	It effectively exploits features in the downsampling process to achieve high-resolution prediction using long-distance residual connectivity.	This model occupies large computing resources, resulting in low training speed and, secondarily, requires pre-trained weights on its backbones.
ResUNet [10]: Extension of standard U-Net model with residual units.	2017	Residual units simplify the training process of deep networks and skip connections could facilitate information propagation.	This model cannot establish the dependency between pixels, which shows poor segmentation performance in blurred images.
Deeplabv3+ [16]: A segmentation network composed of atrous spatial pyramid pooling and decoder modules.	2018	This model can capture sharper object boundaries and extract multi-scale contextual information. Moreover, the resolution of the feature map of the encoder module can be arbitrarily controlled by atrous convolution.	It adopts atrous convolution, which results in the loss of spatially continuous information.
PSPNet [17]: a pyramid scene parsing network embedding context features with PPM.	2017	It can aggregate the context information of different regions to improve the global information extraction ability	It takes a long time to train and performs relatively poorly in detail handling.
CKDNet [18]: a cascade knowledge diffusion network composed of coarse-level segmentation, classification, and fine-level segmentation.	2020	This model adopts knowledge transfer and diffusion strategies to aggregate semantic information from different tasks to boost segmentation performance.	This model consumes a lot of computing resources, resulting in low training speed.
U-Net-FS [19]: A optimal U-Net for free surface segmentation	2020	The outcomes of experiments show U-Net-FS can capture numerous types of free surfaces with a high dice accuracy of over 0.94.	This model can only focus on local information and be trained with a single scale, so it cannot handle the change of image size well.
The fish image segmentation method is combined with the K-means clustering segmentation algorithm and mathematical morphology [20].	2013	The traditional K-means algorithm is improved by using the optimal number of clusters based on the peak number of the image gray histogram.	It is sensitive to the selection of the initial value K and outliers, which will lead to poor segmentation performance.
Transunet [43]: Extension of standard U-Net model with transformer.	2021	This model can extract abundant global context by converting image features into sequences and exploiting the low-level CNN spatial information via a u-shaped architectural design.	This model exploits transposed convolutional layers to restore feature maps. However, it usually results in the chess-board effect, which shows discontinuous predictions among adjacent pixels.
Swin transformer [49]: a hierarchical vision transformer with shifted windows	2021	The shift window scheme improves efficiency via limiting self-attention computation to non-overlapping local windows. Experimental results show SOTA performance in various tasks, including classification, detection, and segmentation tasks.	Constraint by shift window operation. It is time-consuming that this model must be modified and retrained according to different sizes of input.
CoTr [50]: A hybrid network which combines CNN and transformer for 3D medical image segmentation.	2021	It exploits a deformable self-attention mechanism to decrease the spatial and computational complexities of building long-range relationships on multi-scale feature maps.	Since both transformer and 3D volumetric data require a large amount of GPU memory, this method divides the data into small patches and deals with them one at a time, which causes the loss of features from other patches.
TrSeg [51]: A transformer-based segmentation architecture for multi-scale feature extraction.	2021	Different from existing networks for multi-scale feature extraction, this model incorporates a transformer to generate dependencies on original context information, which can adaptively extract multi-scale information well.	This model has relatively few limitations on low-level semantic information extraction.
FAT-Net [52]: Feature adaptive transformer network for skin lesion segmentation.	2021	This model exploits both CNN and transformer encoder branches to extract rich local features and capture the important global context information.	This segmentation network still has limitations when the color change of the image is too complex, and the contrast of the image is too low.

**Table 2 sensors-22-07224-t002:** Hyper-parameters of all the models used in this study.

Parameter	Configuration
Optimizer	SGD
Learning rate	0.03
Weight decay	0.01
Momentum	0.9
Batch size	6
Image size	256*256
Activation function	GELU
Dropout	0.3

**Table 3 sensors-22-07224-t003:** Comparison of various segmentation networks for the marine animal dataset.

Model	*MPA*	*IOU* (Sea Urchin)	*IOU* (Sea Cucumber)	*IOU* (Starfish)	*MIOU*	*Acc*
FCN-8s	36.87%	27.41%	1.68%	15.99%	35.48%	97.05%
U-Net	40.91%	29.14%	3.45%	26.47%	39.01%	96.97%
Attention U-Net	41.54%	30.87%	3.87%	27.28%	39.76%	96.95%
RefineNet	43.14%	32.63%	4.02%	31.32%	41.21%	96.89%
ResUnet	43.79%	33.02%	4.56%	33.29%	41.96%	96.93%
DeepLabv3+	44.24%	34.21%	5.22%	33.23%	42.40%	96.96%
CKDNet	46.05%	39.26%	4.31%	33.58%	43.54%	97.09%
PSPNet	45.38%	37.67%	5.62%	34.13%	43.62%	97.07%
TrSeg	46.32%	40.67%	4.71%	33.47%	43.98%	97.11%
SegFormer [50]	46.67%	39.73%	5.91%	33.51%	44.06%	97.10%
FAT-Net	47.17%	42.97%	5.06%	34.35%	44.87%	97.16%
MulTNet	47.69%	44.14%	5.21%	35.93%	45.63%	97.27%

**Table 4 sensors-22-07224-t004:** Comparison of various segmentation networks for the public ISIC 2018 dataset.

Model	*MPA*	*IOU*	*MIOU*	*Acc*
FCN-8s	89.98%	67.47%	79.03%	92.12%
U-Net	92.19%	76.05%	84.59%	94.36%
Attention U-Net	93.05%	76.35%	84.76%	94.41%
RefineNet	92.85%	76.64%	84.83%	94.32%
ResUnet	93.17%	77.27%	85.37%	94.64%
DeepLabv3+	93.31%	78.21%	85.99%	94.92%
PSPNet	93.08%	79.52%	86.77%	95.15%
CKDNet	93.13%	79.90%	86.99%	95.21%
SegFormer	93.40%	80.19%	87.21%	95.32%
TrSeg	93.22%	80.37%	87.31%	95.33%
FAT-Net	93.65%	81.09%	87.80%	95.56%
MulTNet	94.04%	81.48%	88.09%	95.71%

**Table 5 sensors-22-07224-t005:** Comparison of computational costs.

Model	Total Params	Training Speed (s/Iteration)	Testing Speed (s/Image)
FCN-8s	134 M	0.725	0.291
RefineNet	85 M	0.381	0.148
CKDNet	52 M	0.342	0.135
PSPNet	71 M	0.317	0.124
Segformer	64 M	0.312	0.127
TrSeg	74 M	0.293	0.113
ResUnet	67 M	0.285	0.110
DeepLabv3+	55 M	0.238	0.096
AttentionU-Net	42 M	0.203	0.092
FAT-Net	30 M	0.194	0.089
U-Net	32 M	0.167	0.075
MulTNet	59 M	0.183	0.087

**Table 6 sensors-22-07224-t006:** Ablation experiments.

	ISIC Dataset			Marine Animal Dataset		
Model	*MIOU*	*MPA*	*Acc*	*MIOU*	*MPA*	*Acc*
Transformer	73.89%	89.80%	90.34%	23.99%	25.00%	95.97%
DRCM-Transformer	85.36%	92.29%	94.53%	40.31%	42.33%	96.97%
MulTNet	88.09%	94.04%	95.71%	45.63%	47.69%	97.27%

## Data Availability

We evaluate our network on the public dataset of International Skin Imaging Collaboration (ISIC 2018). The information link is: https://challenge2018.isic-archive.com/ (accessed on 19 October 2021).

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
