# Peer review of "MulTNet: A Multi-Scale Transformer Network for Marine Image Segmentation toward Fishing"

_sensors, 2022, doi:10.3390/s22197224_

Round 1

Reviewer 1 Report

- In this paper, the authors focus on researching a new marine animal segmentation method to enhance the ability of automatic fishing. It is a nice paper, where the authors mention that they will apply its method to the AUVs for the automatic fishing, however they do not explain anything related to AUVs in the whole paper. - Figure 1 (The detailed structure diagram of MulTNet) appears cut on the right side. - In 3.1 (Dimensionality reduction CNN module), the authors mention "..., directly flattening the pixels of input image into a 1D sequence costs large calculated amount." Could the authors explain what the are referring to 1D sequence costs large calculated amount?.

Reviewer 2 Report

In this work, a multi-scale transformer network (MulTNet) for segmenting marine images is proposed. A dimensionality reduction CNN module is presented to reduce the computational time proposed network. In the presented work, progressive down-sampling is designed to fully extract the low-level features. The extracted features are fed into a proposed multiscale transformer module.

The manuscript is interesting; however, the following comments need to be addressed :

Abstract:

1 - Define AUV .

2 – The problem statement should be clearly described

3 – Improvement ratio between the proposed and existing works should be included in the abstract .

Introduction Section :

4 – The organization of the manuscript should be included at the end of the introduction section .

Related Work Section :

5 – For each subsection, include a summary table that includes: 1) reference number and algorithm name, 2) brief methodology, 3) highlights, and 4) limitations .

Proposed Network Section :

6 – The font size of the text in Figure 1 need to be enlarged .

7 – Justification for each part (subsection) in the proposed network requires justification such that it answers how it is work and why it is better than existing .

Experiments and discussion Section :

8 – include reference(s) for the evaluation metrics.

9 – Comparison with recent work is required as well as a thorough discussion .

10 – Computation cost analysis and comparison is missing .

Conclusion Section :

10 – This section is fine. No comments .

Round 2

Reviewer 2 Report

In this work, a multi-scale transformer network (MulTNet) for segmenting marine images is proposed. A dimensionality reduction CNN module is presented to reduce the computational time proposed network. In the presented work, progressive down-sampling is designed to fully extract the low-level features. The extracted features are fed into a proposed multiscale transformer module.

Comments:

The authors have addressed most of the raised comments; however, the following comments need to be addressed :

Abstract:

1 – Please justify why the existing algorithm cannot precisely segment underwater images. This would improve the problem statement.

2 – The included improvement ratio in percentage or normalized?

Introduction Section :

3 – Does other types of image edge detection algorithm affect the segmentation. For example, if orthogonal polynomials used, does this improve the accuracy? You may discuss this in the introduction section. For orthogonal polynomials check doi: 10.1109/ACCESS.2022.3170893   and 10.3390/e23091162 .

Related Work Section :

4 – This section is fine. No comments .

Proposed Network Section :

5 – This section is fine. No comments .

Experiments and discussion Section :

6 – For equation 15, use parenthesis for more clarity.

7 – The computation cost for testing need to be included .

Conclusion Section :

10 – This section is fine. No comments .
